:PLOS | ONE

# Are trials of psychological and psychosocial interventions for schizophrenia and psychosis included in the NICE guidelines pragmatic? A systematic review

Chiara Gastaldon[1]*, Franziska Mosler[2], Sarah Toner[2], Federico Tedeschi[1], Victoria Jane Bird[2], Corrado Barbui[1], Stefan Priebe[2]

**1** WHO Collaborating Centre for Research and Training in Mental Health and Service Evaluation, Department of Neuroscience, Biomedicine and Movement Sciences, Section of Psychiatry, University of Verona, Verona, Italy, **2** Unit for Social and Community Psychiatry, WHO Collaborating Centre for Mental Health Services Development Queen Mary University of London, London, England, United Kingdom

* chiara.gastaldon@gmail.com

**Data Availability Statement:** All relevant data are within the paper and its Supporting Information files.

## Abstract

### Introduction

The NICE clinical guidelines on psychosocial interventions for the treatment of schizophrenia and psychosis in adults are based on the results of randomized controlled trials (RCTs), which may not be studies with a pragmatic design, leading to uncertainty on applicability or recommendations to everyday clinical practice.

### Aim

To assess the level of pragmatism of the evidence used to develop the NICE guideline for psychosocial interventions in psychoses.

### Material and methods

We conducted a systematic and critical appraisal of RCTs used to develop the 'psychological therapy and psychosocial interventions' section of the NICE guideline on the treatment and management of psychosis and schizophrenia in adults, published in 2014. For each study we assessed pragmatism using the pragmatic–explanatory continuum indicator summary-2 (PRECIS-2) and the Cochrane risk of bias tool. The mean score of PRECIS-2, averaging across nine domains, was calculated to describe the level of pragmatism of each individual study.

### Results

A total of 143 studies were included in the analysis. Based on the PRECIS-2 tool, 16.8% were explanatory, 33.6% pragmatic, and 49.7% were rated in an intermediate category. Compared to explanatory studies, pragmatic studies showed a lower risk of bias. Additionally, pragmatism did not significantly improve over time, and no associations were found

**Funding:** The authors received no specific funding for this work.

**Competing interests:** The authors have declared that no competing interests exist.

between pragmatism and a number of trial characteristics. However, studies with a UK leading investigator had the highest mean score of pragmatism. Cognitive behavioural therapy (CBT), art therapy, family intervention, psychoeducation, and adherence therapy, showed the higher average pragmatism scores.

## Conclusions

Two third of studies used to produce NICE recommendations on psychosocial interventions for the treatment of schizophrenia and psychosis in adults are based on studies that did not employ a pragmatic design.

## Introduction

For many years randomized controlled trials (RCTs) have been recognised as the most robust methodology for evaluating the effects of interventions in mental health care and in many other fields of medicine [1]. However, questions have been raised about the generalizability of findings from RCTs to patients treated "in the real world" [2]. This concern is related to the fact that most RCTs are optimised to determine efficacy of interventions in absolute terms, i.e. they are explanatory rather than pragmatic. Pragmatic randomised trials are usually undertaken to help support a decision on whether an intervention should be delivered in a real-world setting. Explanatory randomised trials, by contrast, are undertaken to test whether an intervention is effective under ideal circumstances. There is no simple threshold to determine whether a trial is explanatory or pragmatic, and there are few purely explanatory or pragmatic trials. In most cases, trials include both explanatory and pragmatic characteristics, suggesting a continuum, rather than a dichotomy, between these two polarities [3].

The methodological explanatory rigour of conventional RCT design is seen as unsatisfactory for studying situations of high treatment complexity [4], such as those involved in psychosocial interventions within routine health care [5]. As a consequence, there is general agreement that pragmatic trials should be conducted to show the real-world effectiveness of interventions [6], and thus answer questions of interest to patients, clinicians and policy makers [7].

In the UK, based on the results of RCTs, the National Institute for Health and Care Excellence [8] produces clinical guidelines for health and social care. In 2014 it published the updated guideline on psychological and psychosocial interventions in the treatment of psychosis and schizophrenia in adults [8], which is currently used to guide clinical and policy decisions[9]. A way to understand the applicability to everyday clinical practice of this NICE guideline, the level of pragmatism within the included evidence needs to be assessed. This is especially pertinent for mental health care where the gap between evidence and real-world clinical practice is particularly wide for the treatment of severe mental disorders [10].

In 2009 Thorpe and colleagues [11] defined a tool to help researchers design trials based on whether the purpose of the trial was broadly pragmatic or explanatory: the pragmatic–explanatory continuum indicator summary (PRECIS). In 2015, a new version of PRECIS was developed, improved, and validated with the help of over 80 international trialists, clinicians, and policymakers (PRECIS-2) [3, 12, 13]. Despite being initially developed to inform the design phase of trials, the PRECIS-2 tool has also been applied retrospectively to critically appraise existing trial evidence [14–16].

The aim of this appraisal was therefore to use the PRECIS-2 tool to assess the level of pragmatism of clinical trials testing psychological and psychosocial interventions as included in the

2014 NICE guideline for treatment and management of psychosis and schizophrenia. The appraisal of trials will assist with understanding the evidence-practice gap on psychosocial interventions. Moreover, it aims to identify determinants that affect the pragmatism of studies.

## Materials and methods

### Protocol and registration

The protocol of this work was registered on PROSPERO (CRD42016050116).

### Eligibility criteria

We included all randomized control trials used to develop the 'psychological therapy and psychosocial interventions' section of the NICE guideline on the treatment and management of psychosis and schizophrenia in adults, published in 2014 [8]. Participants included in these trials were 16 years old or older. This section includes trial focused on the following interventions: adherence therapy, art therapies [e.g. music therapy, body psychotherapy), social skills Training (SST), Cognitive Behavioural Therapy (CBT), Family Intervention (FI), Cognitive Remediation (CR), Psychoeducation, Counselling and supportive therapy (CST) and psychodynamic therapy.

### Data collection process

Data extraction was carried out by a one researcher (CG). Two additional researchers (ST, FM) independently assessed the data extracted, the pragmatism and the quality of studies. Any disagreement was discussed with additional authors (VJB, CB, SP). Standardised data collection forms were used to extract data. The information extracted from each study included: year of publication, journal name, country of the lead investigator, sample size, age and diagnosis of participants, type of experimental intervention, results of the study and length of follow-up.

Pragmatism was assessed using the pragmatic–explanatory continuum indicator summary-2 (PRECIS-2). This tool has nine domains (eligibility criteria, recruitment, setting, organization, flexibility (delivery), flexibility (adherence), follow-up, primary outcome, and primary analysis). Each domain can be scored using a 5-point Likert scale in which 1 means very explanatory, 2 rather explanatory, 3 equally pragmatic and explanatory, 4 rather pragmatic and 5 very pragmatic [3]. PRECIS-2 was selected for its reliability and validity [12]. Where it was not possible to give a score to a domain due to a lack of information, the domain was rated as "very explanatory", using a conservative approach, oriented to not overestimate pragmatism.

The risk of bias of each study was assessed using the criteria outlined in the Cochrane Handbook for Systematic Reviews of Interventions for risk of bias [17].

### Statistical analysis

Descriptive statistics (number and percentage of each possible value) were calculated for the following variables: year of publication (published before 1995, between 1995 and 2005 and after 2005); type of journal where the primary reference was published (psychiatric or not psychiatric); multi-centricity (unclear, multi-centric, not multi-centric); country of the lead investigator (UK, Rest of Europe, North America, Asia and Middle East, Australia and unclear); sample size (<50, 50–100, >100 participants, unclear); length of follow-up (post treatment/ end of the intervention; <3 months, 3–6 months, >6 months, unclear); results (intervention significantly better, intervention not better i.e. control better and not significant difference and

unclear); intention-to-treat (ITT) analysis (ITT, not ITT and unclear); diagnosis of partici-pants (only schizophrenia, schizophrenia and other psychoses, unclear); substance abuse in eli-gibility criteria (primary diagnosis of abuse or dependence excluded, secondary diagnosis excluded, users excluded, participants with abuse or dependence not excluded, unclear).

The mean score of PRECIS-2, averaging across nine domains, was calculated to describe the level of pragmatism of each individual study. Studies were then clustered into three groups according to their score: explanatory studies (score< = 2.5), intermediate (score strictly between 2.5 and 3.5) and pragmatic studies (score > = 3.5). These cut-offs where selected to discriminate between studies whose items expressed central values in the explanatory-prag-matic continuum, and those clearly leaning to one side.

Descriptive statistics were reported both for the average pragmatism score (as divided into the categories described above) and for each item on the pragmatism tool, both as continuous and categorical variables, using the cut-off scores described in PRECIS-2 [3]: 1–2 points for explanatory studies, 3 equally pragmatic and explanatory, 4–5 for pragmatic studies.

Additional descriptive and inferential analyses assessed whether pragmatism was associated with selected trial characteristics. First, the mean value of pragmatism for each intervention type was calculated, together with the related confidence intervals. We then analysed the asso-ciation between pragmatism and risk of bias. For each dimension of risk of bias as defined by Higgins and colleagues [17], the percentage of studies classified respectively as "low risk", "high risk" and "unclear" was calculated for each value of pragmatism as a categorical variable. Then, domains of the risk of bias were dichotomized (low risk vs high risk or unclear) and the significance of their association with pragmatism was assessed through Fisher's exact test. Finally, as secondary analyses, we studied whether pragmatism was associated with year of publication, type of journal where the study was published, country of the study's lead investi-gator (as grouped into: UK, Rest of Europe, North America, other), sample size and trial results. We categorized these characteristics as described above. We considered pragmatism both as a continuous variable (by performing ANOVA tests) and as a categorical one (by per-forming Chi-squared tests, or Fisher's exact test, when appropriate).

## Results

### Characteristics of included studies

A total of 143 studies were included in the appraisal, i.e. all studies used to develop recommen-dations described in the NICE guideline for schizophrenia (Fig 1). Included studies assessed the following interventions: adherence therapy, art therapies (e.g. music therapy, body psycho-therapy), social skills Training (SST), Cognitive Behavioural Therapy (CBT), Family Interven-tion (FI), Cognitive Remediation (CR), Psychoeducation (PE), Counselling and supportive therapy (CST) and psychodynamic therapy. Comparators were standard care, wait list control, pharmacological treatment, and other psychological therapies (details and number of studies for each type of intervention are reported in Fig 1).

Table 1 presents the main study characteristics.

Fifty-five studies enrolled only participants with a primary diagnosis of schizophrenia, while forty-nine included participants with related disorders. Only fourteen studies included patients with substance dependence/abuse as a secondary diagnosis. Diagnostic criteria were mainly based on the Diagnostic and Statistical Manual of Mental Disorders Third edition revised or Fourth Edition (DSM-IIIR or IV) or ICD-10. Participants were aged between 18 and 65 years old in the majority of studies. One third of studies was multi-centric; one third were single site studies with classification of the remaining studies unclear. The country of the lead investigator was the UK for 26 studies (18%), North America for 37 studies (of which 35

(27%) USA and 2 Canada), Europe (excluding UK) in 21 studies, other countries in 17 studies, and Australia in four studies (Table 1).

The mean study sample size was 79.0 participants (SD 6.2, CI 66.7 to 91.2). There were 48 studies with a follow-up only at post-treatment, 18 short-term studies with a follow up between 2 and 12 weeks after the end of intervention, 30 medium-term studies with a duration between 13 weeks and 6 months and 34 long-term studies with a follow-up duration of more than 6 months (Table 1).

## Pragmatism of studies

**Descriptive statistics.**   After comparing scores assigned by two researchers, we found an agreement and marked every domain of the PRECIS-2 with a score for each study. We then

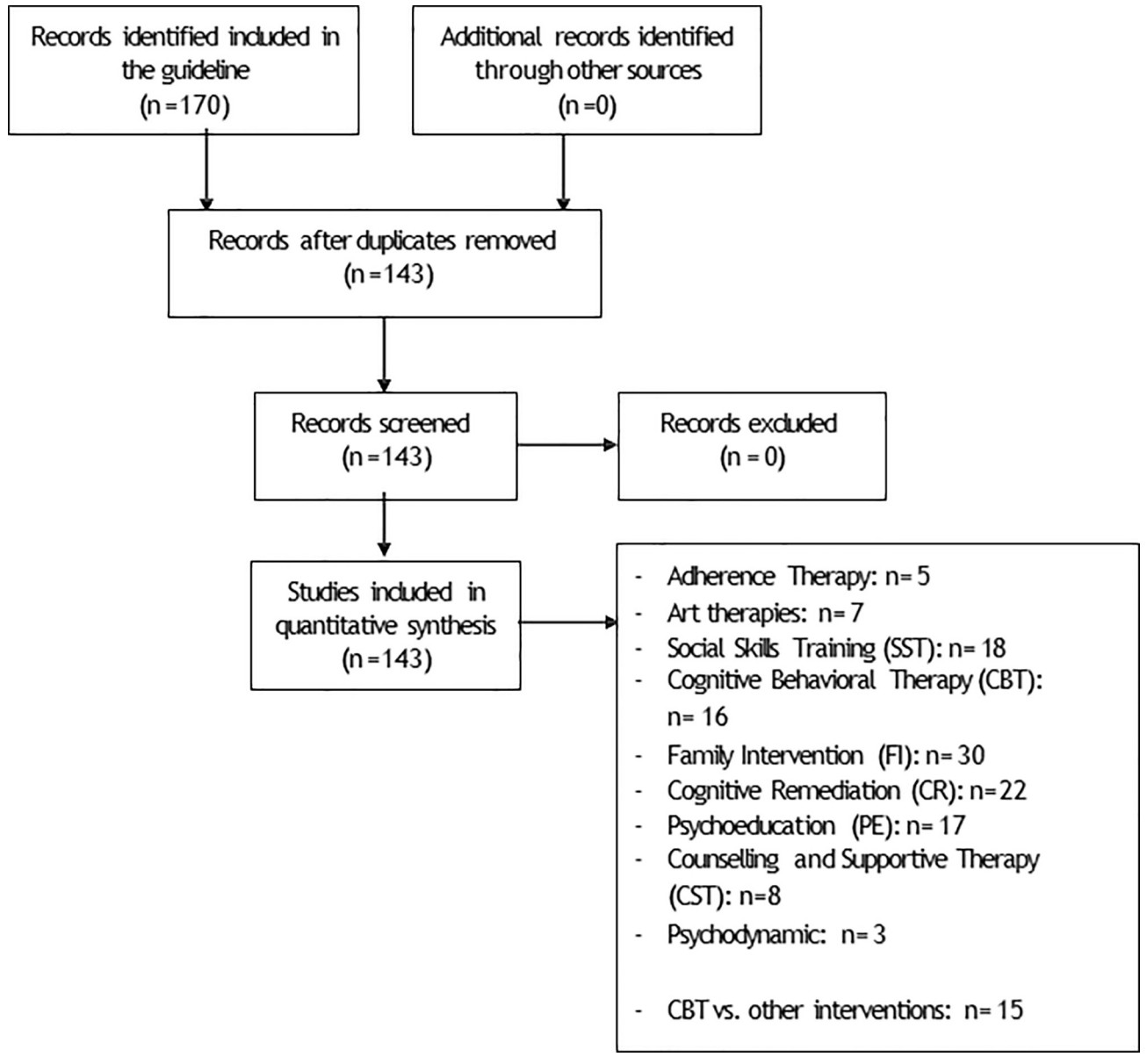

**Fig 1. PRISMA flow chart.**

**Table 1. Characteristics of included studies.**

|  | N of studies | % |
|---|---|---|
| Year of publication: |  |  |
| **Before 1995** | 33 | 23.1 |
| **1995–2005** | 72 | 26.6 |
| **>2005** | 38 | 50.3 |
| Type of journal |  |  |
| **Psychiatric** | 128 | 89.5 |
| **Not psychiatric** | 15 | 10.5 |
| Multicentricity |  |  |
| **Unclear** | 47 | 32.9 |
| **Multicentre** | 43 | 30.1 |
| **Not multicentre** | 53 | 37.1 |
| Country |  |  |
| **Unclear** | 37 | 25.9 |
| **UK** | 26 | 18.2 |
| **North America** | 38 | 26.6 |
| **Rest of Europe** | 21 | 14.7 |
| **Asia and Middle East** | 17 | 11.9 |
| **Australia** | 4 | 2.8 |
| Sample size |  |  |
| **Unclear** | 11 | 7.7 |
| **<50** | 53 | 37.1 |
| **50–100** | 55 | 38.5 |
| **>100** | 24 | 16.8 |
| Length of FOLLOW UP |  |  |
| **Unclear** | 13 | 9.1 |
| **Post treatment** | 48 | 33.6 |
| **<3 months** | 18 | 12.6 |
| **3 months-6months** | 30 | 21.0 |
| **>6 months** | 34 | 23.8 |
| Results |  |  |
| **Unclear** | 14 | 9.8 |
| **Intervention sign. Better** | 89 | 62.2 |
| **Intervention not better** | 40 | 28.0 |
| ITT analysis |  |  |
| **Unclear** | 13 | 9.1 |
| **Yes** | 51 | 35.7 |
| **No** | 79 | 55.2 |
| Diagnosis |  |  |
| **Unclear** | 39 | 27.3 |
| **Schizophrenia** | 55 | 38.5 |
| **Schizophrenia and other psychosis** | 49 | 34.3 |
| Substance abuse in eligibility criteria |  |  |
| **Unclear** | 69 | 48.3 |
| **Primary diagnosis excluded (abuse/dependence)** | 14 | 9.8 |
| **Secondary diagnosis excluded** | 26 | 18.2 |
| **Users excluded** | 12 | 8.4 |
| **Not excluded** | 22 | 15.4 |

**Table 2. Number of studies with an explanatory, intermediate and pragmatic domain.** Mean scores and standard deviations for each domain of the PRECIS-2 tool.

| | Explanatory (n, %) | Intermediate (n, %) | Pragmatic (n, %) | Mean score (SD) |
|---|---|---|---|---|
| 1. **Eligibility** | 61 42.66 | 28 19.58 | 54 37.76 | 3.01 (1.24) |
| 2. **Recruitment** | 61 42.66 | 18 12.59 | 64 44.76 | 2.87 (1.54) |
| 3. **Setting** | 53 37.06 | 20 13.99 | 70 48.95 | 3.27 (1.48) |
| 4. **Organization** | 34 23.78 | 41 28.67 | 68 47.55 | 3.34 (1.23) |
| 5. **Flexibility (Delivery),** | 41 28.67 | 36 25.17 | 66 46.15 | 3.31 (1.26) |
| 6. **Flexibility (Adherence)** | 90 62.94 | 12 8.39 | 41 28.67 | 2.38 (1.58) |
| 7. **Follow-Up** | 47 32.87 | 25 17.48 | 71 49.65 | 3.26 (1.43) |
| 8. **Primary Outcome** | 23 16.08 | 42 29.37 | 78 54.55 | 3.57 (1.05) |
| 9. **Primary Analysis** | 40 27.97 | 34 23.78 | 69 48.25 | 3.48 (1.48) |

calculated and average score for each study. Based on these scores we clustered studies in three categories: 24 (16.8%) were classified as explanatory studies, 71 (49.7%) as intermediate studies and 48 (33.6%) as pragmatic studies. Details of average scores and categories for each study are reported in supplementary material (S1 Appendix).

Moreover, we explored each PRECIS-2 domain: Table 2 presents the distribution of studies in the explanatory, intermediate, or pragmatic categories for each item of the PRECIS-2 tool.

Overall, based on mean scores, the domains rated as most pragmatic were "primary outcome", "primary analysis" and "organization", whereas, "flexibility adherence" and "recruitment" were the most explanatory. Fig 2 shows the average score for each domain on the PRECIS-2 wheel for all included studies. This diagram conveys how pragmatic versus

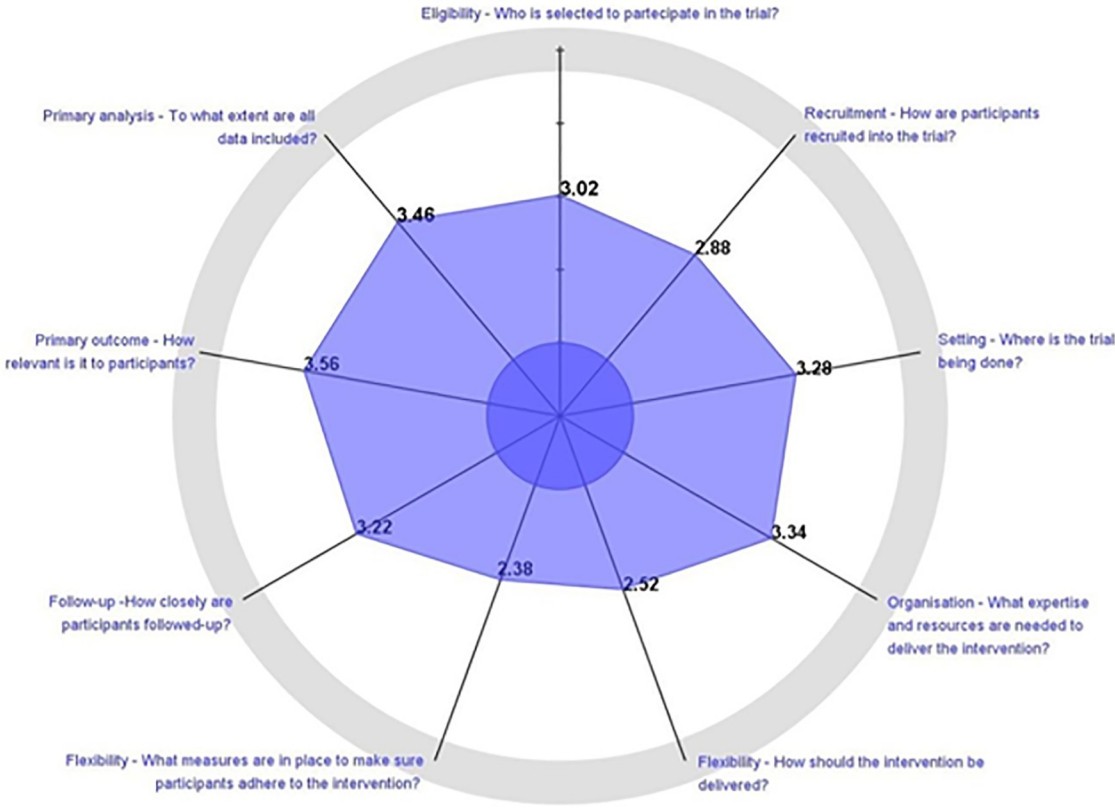

**Fig 2. Cumulative PRECIS Wheel.**

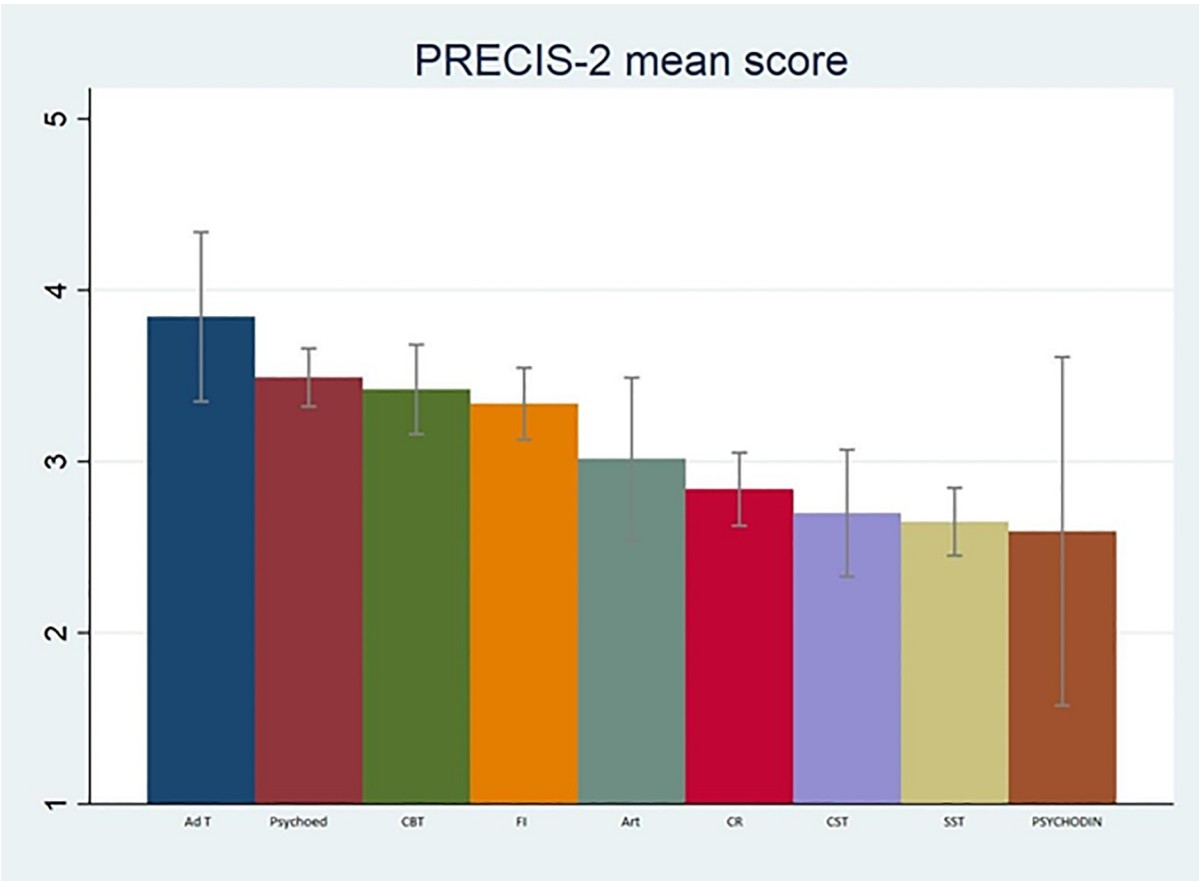

**Fig 3. Mean precis score by intervention (p<0.001).** Ad T = Adherence Therapy (N = 4); PSYCHOED = psychoeducation (N = 17); CBT = Cognitive Behavioural Therapy (N = 33); FI = Family Intervention (N = 31); ART = Art Therapies (N = 7); CR = Cognitive Remediation (N = 22); CST = Counselling and Supportive Therapy (N = 7), SST = Social Skills Training (N = 18); PSYCHODINAM = Psychodynamic and Psychoanalytic Therapies (N = 3)

explanatory a trial is by the distance of the marks on each domain from the centroid: the further away from the centre, the more pragmatic the trial is on that domain.

Pragmatism was additionally studied by type of intervention. Fig 3 shows that CBT, art therapy, family intervention, psychoeducation, and adherence therapy, have the higher average pragmatism scores and this difference was statistically significant (p<0.001).

We assessed the risk of bias for each study and then we reported it in a graphic representation of the quality of studies by pragmatism (Fig 4).

Pragmatic studies show a lower risk of bias in all RoB categories, compared to explanatory and intermediate studies, with the exception of reporting bias. Intermediate studies show an intermediate risk of bias compared to the other two categories. Nevertheless, these differences were statistically significant only for the two categories of selection bias (p< 0.01 in both cases) (More details in the supplementary materials, S1 Table and S1 Fig).

**Relationship between pragmatism and other variables.**   Performing additional descriptive and inferential analyses to assess whether pragmatism was associated with selected trial characteristics we found that pragmatism did not significantly change over time as measured by the year of publication (Table 3).

Similarly, we did not find any significant association between pragmatism and sample size or study results. The analysis evaluating the association between pragmatism and type of

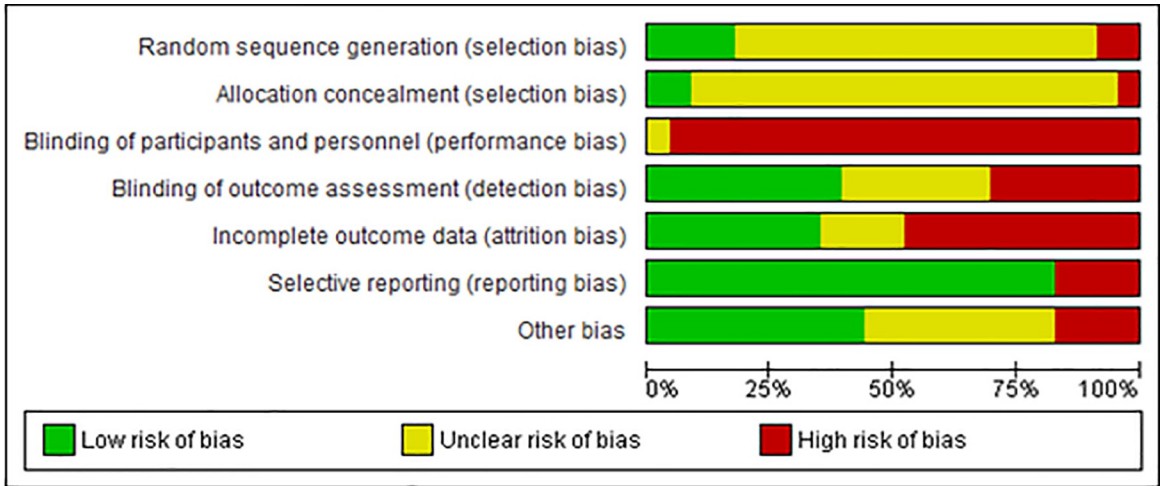

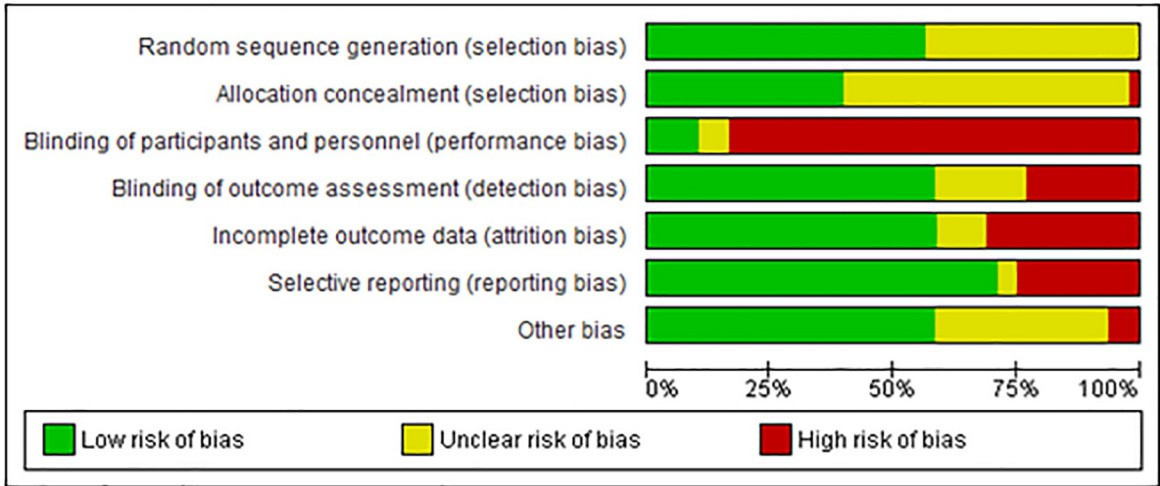

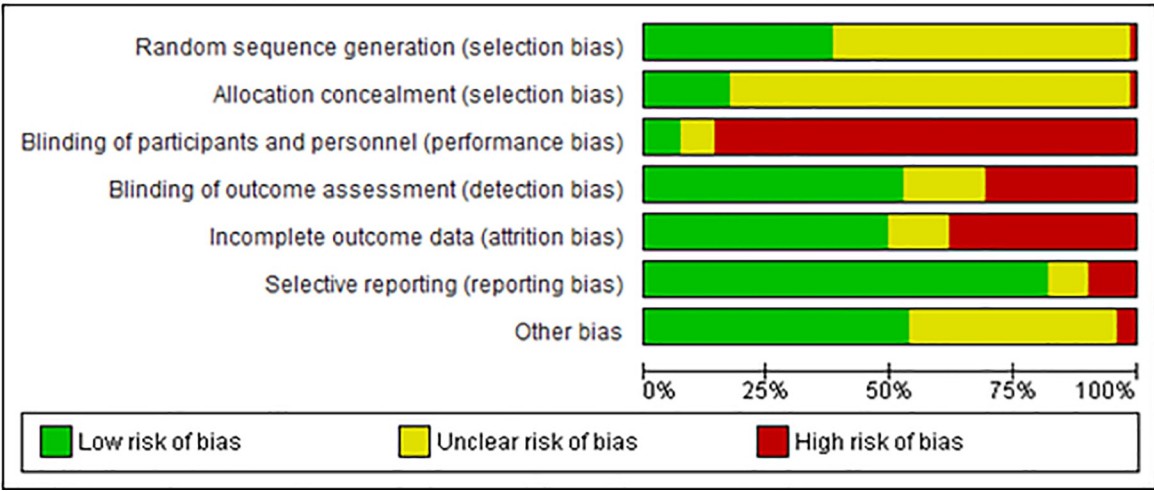

**Fig 4. Risk of bias of studies by PRECIS category.** From the top to the bottom: risk of bias of explanatory, pragmatic and intermediate studies respectively.

journal was not conducted as only 15 out of 143 trials were published in a "non-psychiatric journal". Studies with a UK lead investigator had the highest mean score of pragmatism with a significant association (p<0.001) (Table 3). By replicating the analyses using pragmatism as a three-level categorical variable, the same findings were obtained (more details are reported in table A and B in S2 Table).

## Discussion

### Main findings

This systematic and critical appraisal showed that the 143 psychosocial and psychological trials included in the NICE guideline for schizophrenia and psychosis differed on the extent to which they were rated as pragmatic versus explanatory and consequently on the level of applicability of their results to clinical practice. The average PRECIS-2 score of all studies was between explanatory and intermediate on the continuum.

Across all studies, some domains of the PRECIS-2 tool were rated as more pragmatic then others on average. The PRECIS-2 domains rated as most pragmatic were "primary outcome", "primary analysis" and "organisation", whereas those that were most explanatory were "flexibility-delivery", "flexibility-adherence" and "recruitment". This finding may be just partially explained by the lack of adequate reporting for these domains. These results are consistent with those of Loudon and colleagues [12] who, while testing the discriminant validity and interrater reliability of PRECIS-2, found that these two domains were not statistically better discriminants than chance, as both were poorly described in the trial protocols. Another possible explanation is that it may be easier to be pragmatic for some domains than for others, as pointed out by Johnson and colleagues in their work [18].

Another finding was that trials had different levels of pragmatism depending on the intervention being delivered. Those focusing on art therapy, adherence therapy, psychoeducation,

**Table 3. Analyses of associations.**

|  | Pragmatism mean, SD | Difference: p-value |
|---|---|---|
| **Sample size** |  | **0.294** |
| <50 | 3.1, 0.68 |  |
| 50–100 | 3.2, 0.66 |  |
| >100 | 3.2, 0.64 |  |
| **Year of publication** |  | **0.237** |
| before 1995 | 3.0, 0.62 |  |
| 1995–2005 | 3.2, 0.75 |  |
| after 2005 | 3.2, 0.51 |  |
| **Results** |  | **0.284** |
| Intervention sign. better | 3.1, 0.75 |  |
| Intervention not better | 3.2, 0.65 |  |
| **Country** |  | **<0.001** |
| UK | 3.7, 0.58 |  |
| North America | 2.9, 0.54 |  |
| Europe | 3.2, 0.67 |  |
| Others | 3.2, 0.67 |  |

CBT and family intervention had on average a more pragmatic score, whereas psychodynamic and psychoanalytic therapies, social skills training, cognitive remediation, counselling and supportive therapy had a more explanatory score. This result could be related to the type of intervention itself, as adherence and art therapy, CBT, psychoeducation and family intervention are very flexible interventions, that can be delivered also by non-medical or non-psychological staff (nurses, social workers, other therapists). These results appear to be highly reliable due to the high number of trials with CBT, psychoeducation and family intervention (33, 17 and 31, respectively). This result suggests that the latter interventions may be easier to implement in clinical practice, with implications for policy makers.

The secondary aim of this systematic appraisal was to assess if there is an association between the level of pragmatism on PRECIS-2 and characteristics of the studies. Firstly, explanatory domains did not relate to study quality. By contrast, pragmatic trials had a higher quality than explanatory trials, showing a lower risk of selection bias. This finding is particularly interesting, as it has been suggested that explanatory trials pursue internal validity (quality) at the cost of external validity (pragmatism) [19]. This review does not support this view. We found that pragmatic trials place a premium on external validity while maintaining internal validity, and this is consistent with other previous findings [18, 19].

Reporting bias was the only type of bias in which the risk was lower in explanatory than pragmatic trials. It may be possible that, as explanatory trials choose very specific outcomes at the study design stage in order to assess the efficacy of interventions, they report those outcomes more often than pragmatic trials in order to confirm their initial hypotheses.

As the discussion about the relevance of pragmatism in mental health clinical trials has been raised decades ago and it has been growing progressively [7], we expected to find a trend showing increasing pragmatism over the years. However, the analysis failed to demonstrate this. It appears that pragmatism has not increased from the '80s until 2009, when the psychological and psychosocial chapter of the 2014 guideline was updated.

The only statistically significant association found was between pragmatism and country of the study's lead investigator; UK psychosocial RCTs were on average more pragmatic than studies conducted in other countries. In the UK clinical trials tend to be based in "real world" mental health services that are mostly community based centres with a psychosocial approach. Conversely, in the USA the majority of studies are conducted in high quality academic centres that can be very different from usual care services in staff, resources or type of patients involved.

## Strengths and limits

To our knowledge, this is the first systematic review that analysed the pragmatism of trials included in a guideline for mental health care. So far, most authors have used the PRECIS-2 tool to assess pragmatism at the stage of the trial protocol development, to ensure the design reflects their intended purpose. Within other fields of medicine there are a few examples of PRECIS-2 being used to appraise single studies or studies included in reviews retrospectively [16, 20]. This is relevant as the pragmatism of trials included in the guideline may be an indicator of the overall pragmatism of the guideline itself, i.e. of the applicability of evidence.

These findings may be interpreted with caution because the psychosocial and psychological treatment chapter of the 2014 NICE guideline for psychosis and schizophrenia had not been updated with research evidence published up to 2014 but only to 2009. Since then, more RCTs have been published, and changes in the evidence-base may have been found with new trials potentially being more pragmatic, as interest around this topic has increased in the last 10 years. Despite this issue, the findings of this review remain relevant as current clinical practice in the UK is based on the evidence included in the NICE guideline analysed.

Another limitation was the paucity of details reported in the study reports. This meant that accurate assessments of what actually happened within the trial were difficult, especially in the domains of recruitment and adherence. Instead we assessed the impact of study quality as measured by the Risk of Bias. Finally, although PRECIS-2 is a validated and comprehensive tool, there were difficulties applying it to heterogeneous trials. Due to disagreements with the second and third reviewers, further definition of more detailed "standardized" criteria to ensure internal consistency was developed. A number of researchers from different backgrounds (from both research and clinical practice) and countries were involved in the discussion as suggested by the authors of the PRECIS-2 tool [12].

## Conclusions

Overall, two third of studies used to produce NICE recommendations on psychosocial interventions for the treatment of schizophrenia and psychosis in adults are based on studies that did not employ a pragmatic design.

We would encourage a discussion around weighting and interpreting evidence based on its position on the pragmatism-explanatory continuum when considering new evidence for the next round of updates. As these guidelines currently influence clinical and policy decisions in the UK and in other European countries the results should be considered by clinicians, commissioners, and future research.

Moreover, we have identified several areas of major deficiency in terms of reporting quality and pragmatism including random sequence generation, allocation concealment, explicit outlining of study implementation, blinding, and description of patient recruitment and adherence to therapy. This is in line with the findings of other authors who raised concerns about the methodology employed in the development of NICE guidelines [21, 22]. Recently Dal-Re [14] recommended the inclusion of PRECIS-2 evaluation in trial papers and protocols as this could be useful both to design studies that are more pragmatic and to reduce reporting bias. Based on our findings we would add that future trial reports should improve their comprehensiveness more generally and thereby aid appraisals of quality and pragmatism by using the checklist developed as part of the extension to the CONSORT statement [23, 24]. Further research that is designed for the purpose of pragmatic outcomes and comprehensively reported research is needed in the field of psychosocial intervention for schizophrenia and psychoses to improve the applicability of NICE guidelines within mental health care.

## Supporting information

**S1 Appendix. References of included studies.**
(DOCX)

**S1 Table. Table mean score and category of included studies.**
(DOCX)

**S2 Table. Table A relationship between pragmatism and risk of bias, Table B relationship between pragmatism and other variables.**
(DOCX)

**S1 Fig. Risk of bias of pragmatic studies.** S1B FIG Risk of bias of intermediate studies. S1C FIG Risk of bias of explanatory studies.
(DOCX)

## Author Contributions

**Conceptualization:** Chiara Gastaldon, Victoria Jane Bird, Corrado Barbui, Stefan Priebe.

**Data curation:** Chiara Gastaldon, Franziska Mosler, Sarah Toner.

**Formal analysis:** Chiara Gastaldon, Federico Tedeschi, Corrado Barbui.

**Investigation:** Chiara Gastaldon.

**Methodology:** Corrado Barbui, Stefan Priebe.

**Project administration:** Chiara Gastaldon, Corrado Barbui.

**Software:** Federico Tedeschi.

**Supervision:** Victoria Jane Bird, Corrado Barbui, Stefan Priebe.

**Validation:** Chiara Gastaldon, Federico Tedeschi, Corrado Barbui, Stefan Priebe.

**Visualization:** Chiara Gastaldon.

**Writing – original draft:** Chiara Gastaldon, Corrado Barbui.

**Writing – review & editing:** Chiara Gastaldon, Franziska Mosler, Federico Tedeschi, Victoria Jane Bird, Corrado Barbui, Stefan Priebe.

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
