## [Decision Letter · Decision Letter 0]

24 Jul 2019

PONE-D-19-17507

Are trials of psychological and psychosocial interventions for schizophrenia and psychosis included in the NICE guidelines pragmatic?

PLOS ONE

Dear Dr. CHIARA GASTALDON,

Thank you for submitting your manuscript to PLOS ONE. After careful consideration, we feel that it has merit but does not fully meet PLOS ONE’s publication criteria as it currently stands. Therefore, we invite you to submit a revised version of the manuscript that addresses the points raised during the review process.

ACADEMIC EDITOR: Although it is of interest, the reviewers have raised a number of points which we believe major modifications are necessary to improve the manuscript, taking into account the reviewers' remarks.  

We would appreciate receiving your revised manuscript by Sep 07 2019 11:59PM. To enhance the reproducibility of your results, we recommend that if applicable you deposit your laboratory protocols in protocols.io, where a protocol can be assigned its own identifier (DOI) such that it can be cited independently in the future. For instructions see: http://journals.plos.org/plosone/s/submission-guidelines#loc-laboratory-protocols

We look forward to receiving your revised manuscript.

Kind regards,

Wisit Cheungpasitporn, MD, FACP

University of Mississippi Medical Center

Twitter: @wisit661 Email: wcheungpasitporn@gmail.com 

Academic Editor

PLOS ONE

Journal Requirements:

3. At this time we request that you update the title of this manuscript to identify it as a systematic review, as indicated in the PRISMA checklist.

Reviewers' comments:

Reviewer's Responses to Questions

**Comments to the Author**

1. Is the manuscript technically sound, and do the data support the conclusions?

Reviewer #1: Yes

Reviewer #2: Partly

Reviewer #3: Partly

2. Has the statistical analysis been performed appropriately and rigorously? 

Reviewer #1: Yes

Reviewer #2: Yes

Reviewer #3: I Don't Know

3. Have the authors made all data underlying the findings in their manuscript fully available?

Reviewer #1: Yes

Reviewer #2: Yes

Reviewer #3: Yes

4. Is the manuscript presented in an intelligible fashion and written in standard English?

Reviewer #1: Yes

Reviewer #2: No

Reviewer #3: Yes

5. Review Comments to the Author

Reviewer #1: General aspects

It is a review paper in which he seeks arguments for a treatment guide for schizophrenia and other psychoses. There was a careful inclusion criterion of statistical studies and care, although of basic statistics, the results have discussed aspects that gives support to the conclusion.

Here there are some questions:

1)      The study presents the results of scientific research. Despite being a systematic review study, the presentation of the results is adequate, being a study of the evaluation of NICE pragmatism from previously published works. This point is important for using the guide.

2. Results reported have not been published elsewhere. Although being a review article, the results were organized in an appropriate way to think about the validity or not of the NICE.

3. A basic statistic was used, but the way the authors clustered the data was adequate.

4. Conclusions are presented in an appropriate fashion and are supported by the data.

5. Regarding the references of the 143 studies, will these studies be in supplemental material?

6. There are two initials that don’t have meanings.

7. It was observed that the author does not explain the meaning of two initials – see pg 7.

Reviewer #2: Some revision of the English language is needed. There are some parts of the paper where it is quite difficult to make sense of some sentences. English edit will help to improve the quality of the manuscript. To mention as several as below:

“These cut-offs where selected to distinguish” ???where

“more pragmatic then others” ???then

“other diagnosis” is not correct in grammar

“the risk of bias dimensions were dichotomized” is not correct in grammar.

“both pragmatism” is not correct in grammar

“evidences” is not correct. It is uncountable noun.

“may be interpret” is not correct in grammar.

“involved to stimulate” is not correct in grammar.

Reviewer #3: Definition of pragmatism is unclear. Authors assumed that most people are well versed in the definition and topic of pragmatism in psychiatry. It will be easier for people to understand the message of this paper if definitions of important keywords such as pragmatism and explanatory are well defined.

Authors mentioned that cut off point for studies included was 2009 and NICE was published in 2014, therefore more studies should have been included. However, to honor the protocol of the NICE guidelines, only studies up to 2009 were taken. Therefore it is not a good suggestion to highlight NICE should take more recent studies, as that would violate their protocol. Furthermore, it takes time from the studies taken within the cut off period to the production of a guideline. Furthermore, the authors did say that no statistical significance found between years of publication with pragmatism (page 14 of manuscript).

In page 6 of the manuscript, authors mentioned PRECIS-2 was developed in 2015. NICE guideline was published in 2014, therefore it is not possible to use PRECIS-2 as a measure of pragmatism in including relevant studies for this guidelines, prior to its publication. However it is a good idea to use these criteria retrospectively to examine the level of pragmatism.

Their strength and limitations were addressed.

As for conclusion, in my opinion it is acceptable to say “2/3 of studies used to produce NICE recommendations on psychosocial interventions for the treatment of schizophrenia and psychosis in adults are based on studies that did not employ a pragmatic design”. But to say “This poses serious concerns to the applicability of NICE recommendations” is a rather strong statement as we are aware that studies involving psychosocial treatments are already very scarce in the current picture. Even if these studies were not as pragmatic as indicated by the PRECIS-2, they are still important and relevant in production of the NICE 2014 guideline, which uses an evident-based medicine approach. Perhaps suggestion to use PRECIS-2 as a measure of pragmatism for future studies before including them in future guidelines suggestion is sufficient, without needing to point out the threat that it poses with regards to NICE’s applicability.

6. PLOS authors have the option to publish the peer review history of their article (what does this mean?). If published, this will include your full peer review and any attached files.

Reviewer #1: Yes: Anuska Irene de Alencar

Reviewer #2: No

Reviewer #3: No

---

## [Author Response · Author response to Decision Letter 0]

12 Aug 2019

JOURNAL REQUIREMENTS:

Author: thank you, we changed the file according to the requirements

Author: thank you, we included what you asked

3. At this time we request that you update the title of this manuscript to identify it as a systematic review, as indicated in the PRISMA checklist.

Author: thank you, we changed the title as required by the PRISMA STATEMENT

RESPONSE TO REVIEWERS

Reviewer #1: General aspects

It is a review paper in which he seeks arguments for a treatment guide for schizophrenia and other psychoses. There was a careful inclusion criterion of statistical studies and care, although of basic statistics, the results have discussed aspects that gives support to the conclusion.

Author: thank you, no revision needed

Here there are some questions:

1) The study presents the results of scientific research. Despite being a systematic review study, the presentation of the results is adequate, being a study of the evaluation of NICE pragmatism from previously published works. This point is important for using the guide.

Author: no revision needed

2. Results reported have not been published elsewhere. Although being a review article, the results were organized in an appropriate way to think about the validity or not of the NICE.

Author: no revision needed

3. A basic statistic was used, but the way the authors clustered the data was adequate.

Author: no revision needed

4. Conclusions are presented in an appropriate fashion and are supported by the data.

Author: no revision needed

5. Regarding the references of the 143 studies, will these studies be in supplemental material?

Author: yes studies are already available in supplementary material 

6. There are two initials that don’t have meanings.

Author: Corrected

7. It was observed that the author does not explain the meaning of two initials – see pg 7.

Author: Corrected

Reviewer #2: Some revision of the English language is needed. There are some parts of the paper where it is quite difficult to make sense of some sentences. English edit will help to improve the quality of the manuscript. To mention as several as below:

“These cut-offs where selected to distinguish” ???where

“more pragmatic then others” ???then

“other diagnosis” is not correct in grammar

“the risk of bias dimensions were dichotomized” is not correct in grammar.

“both pragmatism” is not correct in grammar

“evidences” is not correct. It is uncountable noun.

“may be interpret” is not correct in grammar.

“involved to stimulate” is not correct in grammar.

Author: revision of the English language was done, thank you.

Reviewer #3: Definition of pragmatism is unclear. Authors assumed that most people are well versed in the definition and topic of pragmatism in psychiatry. It will be easier for people to understand the message of this paper if definitions of important keywords such as pragmatism and explanatory are well defined.

author: ok, thank you. A definition of pragmatism was added. The new text reads as follows: “Pragmatic randomised trials are usually undertaken to help support a decision on whether an intervention should be delivered in a real-world setting. Explanatory randomised trials, by contrast, are undertaken to test whether an intervention is effective under ideal circumstances. There is no simple threshold to determine whether a trial is explanatory or pragmatic, and there are few purely explanatory or pragmatic trials. In most cases, trials include both explanatory and pragmatic characteristics, suggesting a continuum, rather than a dichotomy, between these two polarities (3).” 

Authors mentioned that cut off point for studies included was 2009 and NICE was published in 2014, therefore more studies should have been included. However, to honor the protocol of the NICE guidelines, only studies up to 2009 were taken. Therefore it is not a good suggestion to highlight NICE should take more recent studies, as that would violate their protocol. Furthermore, it takes time from the studies taken within the cut off period to the production of a guideline. Furthermore, the authors did say that no statistical significance found between years of publication with pragmatism (page 14 of manuscript).

Author: Actually we are not suggesting that NICE should have taken more recent studies in that guideline, but that NICE should update the guideline itself. The guideline is based on evidence published up to 2009. We think that it would be important to update the guideline including newer studies, considering that this field is rapidly evolving. 

In page 6 of the manuscript, authors mentioned PRECIS-2 was developed in 2015. NICE guideline was published in 2014, therefore it is not possible to use PRECIS-2 as a measure of pragmatism in including relevant studies for this guidelines, prior to its publication. However it is a good idea to use these criteria retrospectively to examine the level of pragmatism.

Author: As reported by the referee, and as mentioned in the paper, we retrospectively examined the level of pragmatism of those trials using PRECIS-2. 

Their strength and limitations were addressed.

Author: thank you, no revision needed

As for conclusion, in my opinion it is acceptable to say “2/3 of studies used to produce NICE recommendations on psychosocial interventions for the treatment of schizophrenia and psychosis in adults are based on studies that did not employ a pragmatic design”. But to say “This poses serious concerns to the applicability of NICE recommendations” is a rather strong statement as we are aware that studies involving psychosocial treatments are already very scarce in the current picture. Even if these studies were not as pragmatic as indicated by the PRECIS-2, they are still important and relevant in production of the NICE 2014 guideline, which uses an evident-based medicine approach. Perhaps suggestion to use PRECIS-2 as a measure of pragmatism for future studies before including them in future guidelines suggestion is sufficient, without needing to point out the threat that it poses with regards to NICE’s applicability.

Author: We believe the referee is right. The sentence “This poses serious concerns to the applicability of NICE recommendations” was deleted from the conclusions of the abstract and main text.

---

## [Decision Letter · Decision Letter 1]

10 Sep 2019

[EXSCINDED]

Are trials of psychological and psychosocial interventions for schizophrenia and psychosis included in the NICE guidelines pragmatic? A systematic review

PONE-D-19-17507R1

Dear Dr. CHIARA GASTALDON,

We are pleased to inform you that your manuscript has been judged scientifically suitable for publication and will be formally accepted for publication once it complies with all outstanding technical requirements.

With kind regards,

Wisit Cheungpasitporn, MD, FACP

University of Mississippi Medical Center

Twitter: @wisit661 Email: wcheungpasitporn@gmail.com 

Academic Editor

PLOS ONE

Additional Editor Comments:

I want to commend the authors on their superb efforts to revise the manuscript according to all reviewers’ suggestions. The quality of the manuscript has improved substantially.

Reviewers' comments:

Reviewer's Responses to Questions

**Comments to the Author**

1. If the authors have adequately addressed your comments raised in a previous round of review and you feel that this manuscript is now acceptable for publication, you may indicate that here to bypass the “Comments to the Author” section, enter your conflict of interest statement in the “Confidential to Editor” section, and submit your "Accept" recommendation.

Reviewer #1: All comments have been addressed

Reviewer #2: All comments have been addressed

2. Is the manuscript technically sound, and do the data support the conclusions?

Reviewer #1: Yes

Reviewer #2: Yes

3. Has the statistical analysis been performed appropriately and rigorously? 

Reviewer #1: Yes

Reviewer #2: Yes

4. Have the authors made all data underlying the findings in their manuscript fully available?

Reviewer #1: Yes

Reviewer #2: Yes

5. Is the manuscript presented in an intelligible fashion and written in standard English?

Reviewer #1: Yes

Reviewer #2: Yes

6. Review Comments to the Author

Reviewer #1: It is a review paper in which he seeks arguments for a treatment guide for schizophrenia and other psychoses. There was a careful inclusion criterion of statistical studies and care, although of basic statistics, the results have discussed aspects that gives support to the conclusion.

Reviewer #2: the authors have addressed the raised issues, no further comment. all comments were appropriate and welcome accept as is

7. PLOS authors have the option to publish the peer review history of their article (what does this mean?). If published, this will include your full peer review and any attached files.

Reviewer #1: Yes: Anuska Irene de Alencar

Reviewer #2: No

---

## [Editor Report · Acceptance letter]

13 Sep 2019

PONE-D-19-17507R1 

Are trials of psychological and psychosocial interventions for schizophrenia and psychosis included in the NICE guidelines pragmatic? A systematic review 

Dear Dr. Gastaldon:

I am pleased to inform you that your manuscript has been deemed suitable for publication in PLOS ONE. Congratulations! Your manuscript is now with our production department. 

With kind regards,

on behalf of

Dr. Wisit Cheungpasitporn 

Academic Editor

PLOS ONE